# C–O Coupling of Hydrazones with Diacetyliminoxyl Radical Leading to Azo Oxime Ethers—Novel Antifungal Agents

**DOI:** 10.3390/molecules28237863

**Published:** 2023-11-30

**Authors:** Alexander S. Budnikov, Igor B. Krylov, Mikhail I. Shevchenko, Oleg O. Segida, Andrey V. Lastovko, Anna L. Alekseenko, Alexey I. Ilovaisky, Gennady I. Nikishin, Alexander O. Terent’ev

**Affiliations:** 1N. D. Zelinsky Institute of Organic Chemistry, Russian Academy of Sciences, 47 Leninsky Prospekt, 119991 Moscow, Russia; alsbudnikov@gmail.com (A.S.B.); mishashev4enko@yandex.ru (M.I.S.); segoleg@gmail.com (O.O.S.); ilov@ioc.ac.ru (A.I.I.);; 2All-Russian Research Institute for Phytopathology, B. Vyazyomy, 143050 Moscow, Russia; 3Higher Chemical College of the Russian Academy of Sciences, D. I. Mendeleev University of Chemical Technology of Russia, 9 Miusskaya Square, 125047 Moscow, Russia

**Keywords:** oxime radicals, C–O coupling, fungicidal compounds, crop protection, new modes of action

## Abstract

Selective oxidative C–O coupling of hydrazones with diacetyliminoxyl is demonstrated, in which diacetyliminoxyl plays a dual role. It is an oxidant (hydrogen atom acceptor) and an O-partner for the oxidative coupling. The reaction is completed within 15–30 min at room temperature, is compatible with a broad scope of hydrazones, provides high yields in most cases, and requires no additives, which makes it robust and practical. The proposed reaction leads to the novel structural family of azo compounds, azo oxime ethers, which were discovered to be highly potent fungicides against a broad spectrum of phytopathogenic fungi (*Venturia inaequalis*, *Rhizoctonia solani*, *Fusarium oxysporum*, *Fusarium moniliforme*, *Bipolaris sorokiniana*, *Sclerotinia sclerotiorum*).

## 1. Introduction

The functionalization of organic compounds employing free radicals has emerged as a powerful tool in modern organic chemistry [1,2]. In particular, *N*-oxyl radicals [3] have gained much attention as key agents in oxidative functionalization due to their mild conditions of generation, relatively high stability combined with high reactivity towards organic substrates, and outstanding structural diversity, allowing for control of their properties. However, *N*-oxyl radicals are usually generated *in situ* from corresponding *N*-hydroxy compounds and thus their usage demands oxidants or catalysts and other additives. Frequently, these additional reagents contain transition-metal salts, pose limitations on the substrate scope, and do not correspond to the principles of green chemistry. The peculiar feature of the present work is the use of diacetyliminoxyl [4] as a single ready-to-use free-radical reagent which plays the role of both oxidant and coupling partner for the oxidative functionalization reaction of hydrazones (Figure 1C). Previously, free-radical chemistry of hydrazones was associated mainly with addition and hydrogen substitution reactions of aldehyde hydrazones [5,6,7] (Figure 1A) and cyclizations of hydrazone-derived N-radicals [7,8,9] (Figure 1B). However, ionic mechanisms were proposed for oxidative cyclizations of α,β-unsaturated *N*-tosylhydrazones in some cases [8,10]. It should also be noted that in some functionalizations of type A (Figure 1A), an additional synthetic step of chelate complex formation was necessary for effective radical functionalization of hydrazones [11,12,13]. Hydrazones are reported to undergo peroxidation by *t*-BuOOH in the presence of cobalt–salen complexes with the formation of geminal azoperoxides and geminal azoxyperoxides [14]. Unstable geminal azohydroperoxides [15] are formed as a result of hydrazone autoxidation by molecular oxygen [16,17,18,19]. In general, free-radical functionalization of hydrazones with the formation of azocompounds is less developed compared to methods based on electrophilic attack of hydrazone carbon atoms, such as Michael-type reactions [20,21,22,23], chlorination [24], alkoxylation, or cyanation [25]. Geminal azoacetates are synthesized by the oxidation of hydrazones with Pb(OAc)_4_ [26,27,28]. In the present work (Figure 1C), diacetyliminoxyl was used as the only necessary reagent for high-yielding oxidative C–O coupling with the broad scope of both ketohydrazones and aldehyde-derived hydrazones at room temperature. It should be noted that none of the products of oxidative functionalization of hydrazones mentioned above were considered as fungicidal compounds. Unexpectedly, synthesized C–O coupling products were discovered as a new structural family of fungicides with activity against phytopathogenic fungi at the level of commercially used crop-protection compounds. This finding is very important in the light of the continuous development of strains of phytopathogenic fungi which are resistant against known synthetic fungicide types [29,30,31].

## 2. Results and Discussion

Hydrazone **2aa** was used for the initial experiments with diacetyliminoxyl **1** (Table 1). CH_2_Cl_2_ was used as a solvent because it is a convenient medium for the synthesis and storage of diacetyliminoxyl **1**. The reaction of **2aa** with two equivalents of diacetyliminoxyl under air afforded C–O coupling product **3aa** with an 85% yield (Table 1, entry 1), along with diacetyl oxime **1-H**. The reaction completed in 15 min, as evidenced by the disappearance of the dark red color characteristic of diacetyliminoxyl (for UV-Vis spectrum of **1**, see [32]) and TLC. To check the possible involvement of oxygen as an oxidant [18] in the discovered process, or its possible negative impact on the yield, an experiment under argon was conducted (Table 1, entry 2). However, carrying out the reaction under inert conditions did not lead to a significant change in the yield of **3aa**. The increase in the amount of **1** above the stoichiometric ratio increased the yield of **3aa** by 10% (entry 3 compared to entry 1). Finally, the reaction with excess of hydrazone **2aa** resulted in almost the same yield as in the case of the stoichiometric amount of **2aa** (entry 4).

The conditions of entry 1 of Table 1 were used to test the scope of the discovered C–O coupling (Figure 2). The discovered C–O coupling is compatible with a wide range of hydrazones derived from aromatic ketones (Figure 2A), aromatic aldehydes (Figure 2B), aliphatic ketones (Figure 2C), and aliphatic aldehydes (Figure 2D).

Good yields of C–O coupling products **3aa**–**3ah** (74–96%) were observed for *N*-arylhydrazones of methylarylketones containing electron-donating or electron-withdrawing substituents at benzene rings. The structure of **3ag** was unambiguously confirmed by XRD analysis (see the ESI). The replacement of a methyl group by ethyl did not affect the reaction yield significantly (product **3ai** compared to **3aa**, yields 84–87%). Hydrazone of benzophenone gave an almost quantitative yield of **3aj** despite steric hindrance and the expected low energy of the formed C–O bond [33] due to the steric and electronic effects of phenyl rings. 2-Pyridyl moiety at the nitrogen atom of ketohydrazone was also tolerated (product **3ak**). The reaction took place even in the case of bulky biphenylalkyl hydrazones with long-chain alkyl groups and a 2,4-dinitrophenyl group at the nitrogen atom, albeit with moderate yields of 42–46% (products **3am**, **3an**). The reaction of diacetyliminoxyl with β,γ-unsaturated phenylhydrazone **2al** delivered the C–O coupling product **3al** at 87% with the intact double C=C bond, despite the possible radical cyclization reactions typical of β,γ-unsaturated phenylhydrazones [9]. Moreover, diacetyliminoxyl **1** is known to undergo addition to C=C double bonds at room temperature [34]. Hydrazones derived from aromatic aldehydes also furnish C–O coupling products (**3ba**–**3bd**) in moderate to high yields. Of note, **3bd** was obtained at a 68% yield employing *N*-methyl-substituted hydrazone **2bd**. The reaction proceeded with high yields with acetone phenylhydrazone (product **3ca**), and somewhat lower yields were obtained with higher homologues of acetone (products **3cd**, **3cd**). As in the case of unsaturated hydrazone **2al,** allylacetone phenylhydrazone **2cd** underwent oxidative C–O coupling with the formation of product **3cd** containing intact C=C bond. Cyclic phenylhydrazones with ring sizes of 4–6 furnished the corresponding products **3ce**–**3cg** at a 82–89% yield. The hydrazones of aldehydes reacted smoothly with diacetyliminoxyl, providing azocompounds **3da**–**3de** with a 58–74% yield.

Figure 3 demonstrates the practical applicability of the developed protocol for the synthesis at a 4 mmol scale without chromatographic purification or recrystallization (Figure 3, (1)). Due to the instability of some phenylhydrazones in their pure form, we developed a one-pot procedure delivering the in situ generation of hydrazone that was sequentially added to the solution of diacetyliminoxyl (Figure 3, (2)). Employing this protocol, the corresponding C–O coupling product **3ca** was obtained at a yield of 71%.

Control experiments were conducted to support the plausible reaction mechanism (Figure 4). *N,N*-diphenyl phenylhydrazone **2ao** was introduced in the reaction with diacetyliminoxyl at standard reaction conditions (Figure 4, (1)). There was no C–O coupling product observed by ^1^H-NMR monitoring of the crude reaction mixture after 24 h, indicating that hydrogen atom abstraction from the nitrogen atom is a possible crucial step rather than the addition of an oxime radical at the C=N double bond. The experiment with TEMPO (Figure 4, (2)) is a typical control reaction which is usually employed to intercept possible C-centered radical intermediates. The introduction of two equivalents of TEMPO into the reaction of diacetyliminoxyl **1** with hydrazone **2aa** did not lead to significant changes; product **3aa** was obtained without yield loss (Figure 4, (2)). Moreover, no formation of a TEMPO adduct with C-centered radical was observed (TEMPO recovery 91%), highlighting the exceptionally high efficiency of diacetyliminoxyl in scavenging stabilized C-centered radicals [33].

Two possible reaction pathways can be proposed for the discovered C–O coupling of diacetyliminoxyl with hydrazones (Figure 5). In path I, the hydrogen atom abstraction from hydrazone **2ca** by diacetyliminoxyl **1** is followed by the coupling of the resultant hydrazyl radical **A** with **1**. In path II, diacetyliminoxyl is added to hydrazone **2ca** first, then hydrogen atom from adduct **B** is abstracted. In both cases, the first stage is expected to be rate determining, whereas the second is expected to be very fast or even barrierless. In order to evaluate which path is more plausible, DFT calculations were performed by employing the low-cost but robust B97-3c composite method [35]. The calculations revealed that path I is favored, both kinetically and thermodynamically, compared to path II; however, both pathways demonstrate activation barriers less than 20 kcal·mol^−1^, which are acceptable for room temperature reactions. The fact that path I is energetically more favored than path II is in agreement with the published data on C–O coupling of diacetyliminoxyl with pyrazolones, isoxazolones, and phenols [33]. However, it should be noted that diacetyliminoxyl addition reactions to π-systems were reported recently [34].

## 3. In Vitro Fungicidal Activity of the Synthesized Azo Compounds

In the second part of our research, the synthesized azo oxime ethers **3** were discovered as a new class of fungicides. Fungal diseases of agricultural crops represents one of the major threats to crop production [36,37,38,39]. Phytopathogenic fungi contribute significantly to reductions in crop yield [37,38,39,40] and produce mycotoxins, which can be extremely dangerous food contaminants [41,42,43,44,45,46] (for example, aflatoxins produced by *Aspergillus genus*, trichothecenes by *Fusarium* species, and ergot alkaloids produced by fungi of *Claviceps* genus). Fungicides remain the most effective tool for crop protection against fungal diseases [47]; however, fungicidal resistance development against known active compound classes [29,30,31,48] is a serious threat to crop production, forcing scientists to search for new types of fungicides. Currently, despite the large number of fungicidal compounds used in agriculture, most of them belong to a limited number of classes and share a common mode of action. Namely, succinate dehydrogenase inhibitors (SDHIs), demethylation inhibitors (DMIs, imidazoles and triazoles), quinone outside inhibitors (QoI, or strobilurins), and quinone inside inhibitors (QiI) dominate the fungicide global market and development [49,50,51]. Thus, the discovery of novel antifungal agents with unforeseen modes of action is a primary scientific goal [52,53,54,55,56,57,58]. 

Synthesized products **3** were tested for fungicidal activity at concentrations of 10–30 μg/mL against six phytopathogenic fungi from different taxonomic classes: *V. i.*—*Venturia inaequalis*, *R. s.*—*Rhizoctonia solani*, *F. o.*—*Fusarium oxysporum*, *F. m.*—*Fusarium moniliforme*, *B. s.*—*Bipolaris sorokiniana*, *S. s.*—*Sclerotinia sclerotiorum* (Table 2). Triadimefon and kresoxim-methyl—commercially available fungicides—were used as reference compounds.

As can be seen from Table 2, compounds **3da** and **3dc** exhibit the greatest activity against phytopathogenic fungi. In general, azo oxime ethers with small aliphatic substituents at the quaternary carbon atom (**3ca**–**cg** and **3da**–**de**) possess a higher activity compared to azo oxime ethers bearing aromatic substituents at the quaternary center (**3aa**, **3ae**, **3ag**, and **3ba**). Compounds **3aj**, **3al**, **3am**, and **3an** with bulky substituents at quaternary carbon atom do not show significant fungicidal activity, as well as azo oxime ether **3bd** with a Me substituent at the nitrogen atom. Aldehyde-derived azo oxime ethers (**3ba** and **3da**–**de**), in general, are superior to ketone derivatives (**3aa**, **3ae**, **3ag**, and **3ca**–**cg**). In the series of long-chain alkyl or cyclic azo compounds, activity decreases with increasing alkyl chain (**3ca**, **3cb**, **3cc**) or ring size (**3ce**, **3cf**, **3cg**). AIBN, an alkyl azo derivative frequently used as a radical initiator taken for comparison, does not show significant activity. Compounds with a diacetyl oxime moiety, obtained by oxidative C–O coupling of diacetyliminoxyl with alkenes [34], pyrazolones [33], phenols [33], and dicarbonyl compounds [59], were also tested for fungicidal activity. None of them show essential mycelium growth inhibition, indicating that diacetyl oxime moiety itself is not sufficient for the manifestation of the observed fungicidal activity. It is noteworthy that the activity of the synthesized azo compounds in the present study was not predictable due to their structural novelty. The closest related fungicidal compounds are generally diaryl azo derivatives [60,61,62] and substituted oxime derivatives with a RO–N=C–N=N-Ar fragment at the oxime moiety [63]. In contrast to these fungicides, the azo oxime ethers reported in the present work contain a tertiary C(sp^3^) atom at the azo group. The activity of the azo oxime ethers **3ca**, **3cd**–**cf**, and **3da**–**de** is comparable to that of triadimefon and kresoxim-methyl, which are commercially available fungicides widely used in crop protection.

EC_50_ values were measured for the most promising azo compounds, **3ca** and **3da**, and reference compound kresoxim-methyl (Table 3).

Synthesized azo compounds **3ca** and **3da** have a similar activity spectrum that greatly differs from that of kresoxim-methyl. Overall, the EC_50_ values of **3ca** and **3da** are comparable to those of kresoxim-methyl; however, at higher concentrations, these azo compounds demonstrate stronger mycelium growth inhibition (Table 2).

## 4. Materials and Methods

In all experiments RT stands for 22–25 °C. ^1^H and ^13^C NMR spectra were recorded on Bruker AVANCE II 300 and Bruker Fourier 300HD (300.13 for ^1^H and 75.47 MHz for ^13^C, respectively) spectrometers in CDCl_3_. Chemical shifts were reported in parts per million (ppm), and the residual solvent peak was used as an internal reference: ^1^H (CDCl_3_ δ = 7.26 ppm); ^13^C (CDCl_3_ δ = 77.16 ppm). Multiplicity was indicated as follows: s (singlet), d (doublet), t (triplet), q (quartet), m (multiplet). Coupling constants were reported in Hertz (Hz). FT-IR spectra were recorded on Bruker Alpha instrument. High resolution mass spectra (HR-MS) were measured on a Bruker maXis instrument using electrospray ionization (ESI). The measurements were performed in a positive ion mode (interface capillary voltage—4500 V); mass range from *m/z* 50 to *m/z* 3000 Da; external calibration with Electrospray Calibrant Solution (Fluka). A syringe injection was used for all acetonitrile solutions (flow rate 3 µL/min). Nitrogen was applied as a dry gas; interface temperature was set at 180 °C.

Phenylhydrazine 97%, *p*-tolylhydrazine hydrochloride 98%, 4-chlorophenylhydrazine hydrochloride 97%, 4-(trifluoromethyl) phenylhydrazine 96%, 2-hydrazinopyridine 98%, 2,4-dinitrophenylhydrazine 97%, methylhydrazine 98%, acetophenone 98%, 4-methylacetophenone 95%, 4-nitroacetophenone 98%, 4-methoxyacetophenone 98%, 4-bromoacetophenone 98%, 2-hydroxyacetophenone 98%, propiophenone 99%, benzophenone 99%, benzaldehyde 98%, 4-chlorobenzaldehyde 98%, 4-methoxybenzaldehyde 99%, 4-heptanone 98%, 6-undecanone 97%, 5-hexen-2-one 98%, cyclobutanone 98%, cyclopentanone 99%, cyclohexanone 99%, acetaldehyde 99.5%, propionaldehyde 98%, isobutyraldehyde 99+%, pivaldehyde 96%, hexanal 96%. Hydrazones **2** were synthesized by condensation with the corresponding carbonyl compounds [64,65,66,67,68,69]. Ketones and corresponding hydrazones **2l**–**n** were synthesized according to published procedures [70,71,72,73]. Compounds **4** and **8** [34], **5** [4,33], **7** [33], and **6** [59] with a diacetyl oxime moiety were synthesized by oxidative C–O coupling according to published procedures. CH_2_Cl_2_ was distilled prior to use. Acetone was distilled over KMnO_4_. The preparation of diacetyliminoxyl radical is described earlier in [4]. Then, Pb(OAc)_4_ (469 mg, 1.0 mmol) was added to a stirred solution of diacetyl oxime (258 mg, 2 mmol) in CH_2_Cl_2_ (4 mL) with vigorous stirring. Stirring was continued for 10 min; then, the reaction mixture was chromatographed on silica gel using CH_2_Cl_2_ as eluent. The fraction corresponding to the dark-red spot was collected, so that the volume of the fraction was 50 mL.


**General reaction conditions for Table 1**


Hydrazone **2aa** (1–2 mmol) was added to a stirred solution of diacetyliminoxyl radical **1** (2–3 mmol) in CH_2_Cl_2_ (50 mL), prepared as described earlier in [4], at room temperature. The reaction mixture was stirred for 15 min under air (entries 1, 3, 4) or under argon (entry 2) atmosphere, until the dark red color of diacetyliminoxyl disappeared. After that, the reaction mixture was rotary evaporated under a water-jet vacuum. Yields were determined by ^1^H NMR using 1,1,2,2-tetrachloroethane as an internal standard.


**Experimental details for Figure 2**


Hydrazone **2** (1 mmol, 134–460 mg) was added to a stirred solution of diacetyliminoxyl **1** (2 mmol) in CH_2_Cl_2_ (50 mL) at room temperature. The reaction mixture was stirred at RT for 15–30 min until the red color of diacetyliminoxyl disappeared. After that, the reaction mixture was rotary evaporated under a water-jet vacuum. C–O coupling products **3** were isolated by column chromatography on silica gel.

**(*E*)-3-((1-phenyl-1-(phenyldiazenyl)ethoxy)imino)pentane-2,4-dione**, **3aa,** was synthesized as a yellow oil (84%, purified by column chromatography with DCM as eluent). **^1^H NMR** (300.13 MHz, CDCl_3_): δ = 7.81–7.69 (m, 2H), 7.56–7.43 (m, 5H), 7.43–7.28 (m, 3H), 2.48 (s, 3H), 2.27 (s, 3H), 2.05 (s, 3H). **^13^C NMR** (75.47 MHz, CDCl_3_): δ = 198.8, 194.7, 156.6, 151.6, 139.4, 131.6, 129.2, 128.7, 128.6, 126.5, 122.9, 105.2, 30.5, 25.9, 23.8. **FT-IR** (thin layer): ν_max_ = 1725, 1690, 1363, 960, 695. **HR-MS** (ESI): *m/z* = 360.1313, calcd. for C_19_H_19_N_3_O_3_+Na^+^: 360.1319.**(*E*)-3-((1-(phenyldiazenyl)-1-(*p*-tolyl)ethoxy)imino)pentane-2,4-dione**, **3ab,** was synthesized as a yellow oil (96%, purified by column chromatography with DCM as eluent). **^1^H NMR** (300.13 MHz, CDCl_3_): δ = 7.81–7.71 (m, 2H), 7.54–7.44 (m, 3H), 7.41 (d, *J* = 8.2 Hz, 2H), 7.19 (d, *J* = 8.2 Hz, 2H), 2.49 (s, 3H), 2.35 (s, 3H), 2.29 (s, 3H), 2.06 (s, 3H). **^13^C NMR** (75.47 MHz, CDCl_3_): δ = 198.7, 194.7, 156.5, 151.6, 138.6, 136.4, 131.4, 129.24, 129.18, 126.5, 122.8, 105.2, 30.5, 25.8, 23.5, 21.2. **FT-IR** (thin layer): ν_max_ = 1725, 1687, 1363, 1305, 968. **HR-MS** (ESI): *m/z* = 352.1654, calcd. for C_20_H_21_N_3_O_3_+H^+^: 352.1656.**(*E*)-3-((1-((4-chlorophenyl)diazenyl)-1-(*p*-tolyl)ethoxy)imino)pentane-2,4-dione**, **3ac,** was synthesized as a yellow oil (81%, purified by column chromatography with DCM as eluent). **^1^H NMR** (300.13 MHz, CDCl_3_): δ = 7.70 (d, *J* = 8.7 Hz, 2H), 7.44 (d, *J* = 8.7 Hz, 2H), 7.39 (d, *J* = 8.3 Hz, 2H), 7.19 (d, *J* = 8.3 Hz, 2H), 2.47 (s, 3H), 2.35 (s, 3H), 2.29 (s, 3H), 2.04 (s, 3H). **^13^C NMR** (75.47 MHz, CDCl_3_): δ = 198.6, 194.6, 156.7, 149.9, 138.7, 137.5, 136.2, 129.4, 129.3, 126.4, 124.1, 105.3, 30.5, 25.8, 23.4, 21.2. **FT-IR** (thin layer): ν_max_ = 1726, 1690, 1362, 1300, 1088, 959. **HR-MS** (ESI): *m/z* = 386.1252, 388.1230, calcd. for C_20_H_20_ClN_3_O_3_+H^+^: 386.1266, 388.1238.**(*E*)-3-((1-phenyl-1-((4-(trifluoromethyl)phenyl)diazenyl)ethoxy)imino)pentane-2,4-dione**, **3ad,** was synthesized as a yellow oil (95%, purified by column chromatography with DCM as eluent). **^1^H NMR** (300.13 MHz, CDCl_3_): δ = 7.85 (d, *J* = 8.4 Hz, 2H), 7.75 (d, *J* = 8.4 Hz, 2H), 7.58–7.49 (m, 2H), 7.47–7.32 (m, 3H), 2.49 (s, 3H), 2.29 (s, 3H), 2.10 (s, 3H). **^13^C NMR** (75.47 MHz, CDCl_3_): δ = 198.5, 194.5, 156.7, 153.3, 138.8, 132.89 (q, *J* = 32.7 Hz), 128.9, 128.7, 126.42 (q, *J* = 3.6 Hz), 123.84 (q, *J* = 272.5 Hz), 123.0, 105.5, 30.5, 25.7, 23.6. **FT-IR** (thin layer): ν_max_ = 1726, 1692, 1364, 1324, 1169, 1131, 1066, 959. **HR-MS** (ESI): *m/z* = 428.1198, calcd. for C_20_H_18_F_3_N_3_O_3_+Na^+^: 428.1192.**(*E*)-3-((1-(4-nitrophenyl)-1-(phenyldiazenyl)ethoxy)imino)pentane-2,4-dione**, **3ae,** was synthesized as a yellow oil (98%, purified by column chromatography with DCM as eluent). **^1^H NMR** (300.13 MHz, CDCl_3_): δ = 8.25 (d, *J* = 8.9 Hz, 2H), 7.81–7.67 (m, 4H), 7.56–7.45 (m, 3H), 2.48 (s, 3H), 2.24 (s, 3H), 2.03 (s, 3H). **^13^C NMR** (75.47 MHz, CDCl_3_): δ = 198.2, 194.3, 157.1, 151.3, 148.0, 146.6, 132.2, 129.4, 127.8, 123.8, 123.0, 104.1, 30.4, 25.9, 24.4. **FT-IR** (thin layer): ν_max_ = 1727, 1693, 1605, 1522, 1350, 1300, 1142, 1109, 1079, 1067, 958, 855, 769, 758, 693. **HR-MS** (ESI): *m/z* = 405.1161, calcd. for C_19_H_18_N_4_O_5_+Na^+^: 405.1169.**(*E*)-3-((1-(4-methoxyphenyl)-1-(phenyldiazenyl)ethoxy)imino)pentane-2,4-dione**, **3af,** was synthesized as a yellow oil (75%, purified by column chromatography with DCM as eluent). **^1^H NMR** (300.13 MHz, CDCl_3_): δ = 7.80–7.67 (m, 2H), 7.55–7.38 (m, 5H), 6.95–6.82 (m, 2H), 3.80 (s, 3H), 2.47 (s, 3H), 2.28 (s, 3H), 2.04 (s, 3H). **^13^C NMR** (75.47 MHz, CDCl_3_): δ = 198.8, 194.7, 159.9, 156.5, 151.6, 131.4, 129.2, 128.0, 122.8, 113.9, 105.1, 55.4, 30.5, 25.8, 23.3. **FT-IR** (thin layer): ν_max_ = 1725, 1690, 1608, 1514, 1363, 1303, 1253, 1185, 1109, 1030, 960, 834, 769. **HR-MS** (ESI): *m/z* = 390.1423, calcd. for C_20_H_21_N_3_O_4_+Na^+^: 390.1424.**(*E*)-3-((1-(4-bromophenyl)-1-(phenyldiazenyl)ethoxy)imino)pentane-2,4-dione**, **3ag,** was synthesized as yellow crystals (82%, purified by column chromatography with DCM as eluent). Mp = 90–91 °C. **^1^H NMR** (300.13 MHz, CDCl_3_): δ = 7.81–7.70 (m, 2H), 7.55–7.45 (m, 5H), 7.44–7.35 (m, 2H), 2.47 (s, 3H), 2.27 (s, 3H), 2.01 (s, 3H). **^13^C NMR** (75.47 MHz, CDCl_3_): δ = 198.6, 194.6, 156.8, 151.4, 138.6, 131.8, 129.3, 128.4, 123.1, 122.9, 104.6, 102.8, 30.5, 25.9, 23.8. **FT-IR** (thin layer): ν_max_ = 1773, 1484, 1397, 1362, 1302, 1135, 1078, 1010, 966, 920, 828, 685, 550. **HR-MS** (ESI): *m/z* = 416.0608, 418.0592, calcd. for C_19_H_18_BrN_3_O_3_+H^+^: 416.0604, 418.0585. Single crystal X-ray analysis is available (see Appendix A).**(*E*)-3-((1-(2-hydroxyphenyl)-1-(phenyldiazenyl)ethoxy)imino)pentane-2,4-dione**, **3ah,** was synthesized as a pale yellow solid (74%, purified by column chromatography with DCM as eluent) Mp = 103–104 °C. **^1^H NMR** (300.13 MHz, CDCl_3_): δ = 8.16 (s, 1H), 7.74–7.71 (m, 2H), 7.54–7.49 (m, 3H), 7.36–7.27 (m, 2H), 6.97–6.90 (m, 2H), 2.46 (s, 3H), 2.25 (s, 3H), 2.09 (s, 3H). **^13^C NMR** (75.47 MHz, CDCl_3_): δ = 198.2, 194.4, 157.0, 155.4, 150.9, 132.5, 131.3, 129.6, 127.3, 124.0, 123.0, 120.3, 118.5, 106.9, 30.6, 25.9, 22.8. **FT-IR** (thin layer): ν_max_ = 1727, 1692, 1483, 1458, 1364, 1299, 1246, 1201, 1105, 957, 939, 76. **HR-MS** (ESI): *m/z* = 376.1260, cald. for C_19_H_19_N_3_O_4_+Na^+^ = 376.1268.**(*E*)-3-((1-phenyl-1-(phenyldiazenyl)propoxy)imino)pentane-2,4-dione**, **3ai,** was synthesized as a yellow oil (87%, purified by column chromatography with DCM as eluent). **^1^H NMR** (300.13 MHz, CDCl_3_): δ = 7.85–7.69 (m, 2H), 7.57–7.53 (m, 2H), 7.51–7.47 (m, 3H), 7.44–7.30 (m, 3H), 2.52 (s, 3H), 2.60–2.35 (m, 2H), 2.23 (s, 3H), 0.88 (t, *J* = 7.4 Hz, 3H). **^13^C NMR** (75.47 MHz, CDCl_3_): δ = 198.8, 194.7, 156.7, 151.6, 138.2, 131.4, 129.2, 128.5, 128.3, 126.9, 122.8, 106.9, 31.1, 30.3, 25.8, 7.7. **FT-IR** (thin layer): ν_max_ =2979, 1726, 1691, 1450, 1363, 1296, 1138, 1070, 963, 763, 699, 691. **HR-MS** (ESI): *m/z* = 374.1472, calcd. for C_20_H_21_N_3_O_3_+Na^+^: 374.1475.**(*E*)-3-((diphenyl(phenyldiazenyl)methoxy)imino)pentane-2,4-dione**, **3aj,** was synthesized as a slightly yellow solid (98%, purified by column chromatography with DCM as eluent). Mp = 103–104 °C. **^1^H NMR** (300.13 MHz, CDCl_3_): δ = 7.86–7.75 (m, 2H), 7.58–7.45 (m, 7H), 7.43–7.30 (m, 6H), 2.56 (s, 3H), 2.05 (s, 3H). **^13^C NMR** (75.47 MHz, CDCl_3_): δ = 198.7, 194.7, 156.3, 151.5, 139.9, 131.6, 129.3, 128.6, 128.4, 128.0, 123.0, 105.6, 30.1, 25.7. **FT-IR** (thin layer): ν_max_ = 1725, 1686, 1300, 1013, 976, 941, 762, 695. **HR-MS** (ESI): *m/z* = 422.1461, calcd. for C_24_H_21_N_3_O_3_+Na^+^: 422.1475.**(*E*)-3-((1-phenyl-1-(pyridin-2-yldiazenyl)ethoxy)imino)pentane-2,4-dione**, **3ak,** was synthesized as a yellow oil (89%, purified by column chromatography with PE/EtOAc = 2/5 as eluent). **^1^H NMR** (300.13 MHz, CDCl_3_): δ = 8.70 (d, *J* = 4.2 Hz, 1H), 7.85 (td, *J* = 7.7, 1.8 Hz, 1H), 7.60–7.48 (m, 3H), 7.46–7.29 (m, 4H), 2.48 (s, 3H), 2.28 (s, 3H), 2.12 (s, 3H). **^13^C NMR** (75.47 MHz, CDCl_3_): δ = 198.6, 194.6, 162.2, 156.8, 149.6, 138.6, 138.5, 129.0, 128.7, 126.5, 125.8, 114.3, 106.0, 30.6, 25.9, 23.5. **FT-IR** (thin layer): ν_max_ = 1725, 1690, 1583, 1455, 1425, 1363, 1299, 1261, 1194, 1145, 1119,1069, 955, 791, 770, 699. **HR-MS** (ESI): *m/z* = 339.1448, calcd. for C_18_H_18_N_4_O_3_+H^+^: 339.1452.**3-((((*Z*)-1,4-diphenyl-1-((*E*)-phenyldiazenyl)but-3-en-1-yl)oxy)imino)pentane-2,4-dione**, **3al,** was synthesized as a slightly yellow viscous gum (87%, purified by column chromatography with DCM as eluent). **^1^H NMR** (300.13 MHz, CDCl_3_): δ = 7.79–7.70 (m, 2H), 7.58–7.45 (m, 5H), 7.45–7.17 (m, 8H), 6.54 (d, *J* = 11.8 Hz, 1H), 5.58 (dt, *J* = 11.8, 7.2 Hz, 1H), 3.57 (dd, *J* = 7.2, 1.8 Hz, 2H), 2.50 (s, 3H), 1.98 (s, 3H). **^13^C NMR** (75.47 MHz, CDCl_3_): δ = 198.6, 194.7, 156.6, 151.5, 137.3, 137.2, 132.5, 131.6, 129.2, 128.74, 128.67, 128.61, 128.4, 127.0, 126.9, 124.9, 122.9, 106.3, 36.2, 30.3, 25.6. **FT-IR** (thin layer): ν_max_ = 1725, 1690, 1600, 1494, 1449, 1363, 1301, 1193, 1059, 1018, 1003, 950, 765, 699. **HR-MS** (ESI): *m/z* = 462.1781, calcd. for C_27_H_25_N_3_O_3_+Na^+^: 462.1788.**(*E*)-3-(((1-([1,1′-biphenyl]-4-yl)-1-((2,4-dinitrophenyl)diazenyl)hexyl)oxy)imino)pentane-2,4-dione**, **3am,** was synthesized as a viscous orange gum (46%, purified by column chromatography with DCM as eluent). **^1^H NMR** (300.13 MHz, CDCl_3_): δ = 8.83 (d, *J* = 2.3 Hz, 1H), 8.51 (dd, *J* = 8.7, 2.3 Hz, 1H), 7.73–7.56 (m, 6H), 7.52–7.41 (m, 3H), 7.41–7.32 (m, 1H), 2.60–2.47 (m, 2H), 2.44 (s, 3H), 2.33 (s, 3H), 1.42–1.20 (m, 6H), 0.94–0.77 (m, 3H). **^13^C NMR** (75.47 MHz, CDCl_3_): δ = 198.5, 194.4, 157.1, 148.6, 148.1, 146.1, 141.7, 140.2, 135.6, 129.0, 128.4, 127.9, 127.5, 127.18, 127.15, 120.5, 120.3, 108.1, 37.3, 31.8, 30.2, 25.8, 22.6, 22.4, 14.0. **FT-IR** (thin layer): ν_max_ = 3103, 2957, 2931, 2869, 1726, 1692, 1608, 1536, 1487, 1346, 1298, 1147, 954, 836, 766, 744, 698. **HR-MS** (ESI): *m/z* = 582.1955, calcd. for C_29_H_29_N_5_O_7_+Na^+^: 582.1959.**(*E*)-3-(((1-([1,1′-biphenyl]-4-yl)-1-((2,4-dinitrophenyl)diazenyl)octyl)oxy)imino)pentane-2,4-dione**, **3an,** was synthesized as a viscous orange gum (42%, purified by column chromatography with DCM as eluent). **^1^H NMR** (300.13 MHz, CDCl_3_): δ = 8.82 (d, *J* = 2.3 Hz, 1H), 8.51 (dd, *J* = 8.7, 2.3 Hz, 1H), 7.78–7.55 (m, 6H), 7.54–7.40 (m, 3H), 7.39–7.29 (m, 1H), 2.66–2.49 (m, 2H), 2.45 (s, 3H), 2.34 (s, 3H), 1.55–1.09 (m, 10H), 0.86 (t, *J* = 6.5 Hz, 3H). **^13^C NMR** (75.47 MHz, CDCl_3_): δ = 198.5, 194.4, 157.1, 148.6, 148.1, 146.1, 141.8, 140.2, 135.6, 129.0, 128.4, 127.9, 127.5, 127.2, 120.5, 120.3, 108.1, 37.4, 31.8, 30.3, 29.6, 29.1, 25.8, 23.0, 22.7, 14.2. **FT-IR** (thin layer): ν_max_ = 1724, 1691, 1607, 1545, 1541, 1346, 1297, 1194, 1146, 963, 835, 766, 747, 698. **HR-MS** (ESI): *m/z* = 605.2712, calcd. for C_31_H_33_N_5_O_7_+H^+^: 605.2718.**(*E*)-3-((phenyl(phenyldiazenyl)methoxy)imino)pentane-2,4-dione**, **3ba,** was synthesized as a yellow oil (79%, purified by column chromatography with DCM as eluent). **^1^H NMR** (300.13 MHz, CDCl_3_): δ = 7.79–7.76 (m, 2H), 7.55–7.51 (m, 2H), 7.50–7.47 (m, 3H), 7.44–7.39 (m, 3H), 6.50 (s, 1H), 2.46 (s, 3H), 2.35 (s, 3H). **^13^C NMR** (75.47 MHz, CDCl_3_): δ = 198.9, 194.4, 157.2, 151.6, 134.4, 131.9, 130.7, 129.8, 129.3, 129.0, 127.9, 123.7, 107.3, 30.6, 25.9. **FT-IR** (thin layer): ν_max_ =1725, 1693, 1453, 1419, 1360, 1195, 1098, 1019, 952, 766, 695. **HR-MS** (ESI): *m/z* = 346.1162 cald. for C_18_H_17_N_3_O_3_+Na^+^ = 346.1162.**(*E*)-3-(((4-chlorophenyl)(phenyldiazenyl)methoxy)imino)pentane-2,4-dione**, **3bb,** was synthesized as a yellow powder (58%, purified by column chromatography with DCM as eluent). Mp = 49–50 °C. **^1^H NMR** (300.13 MHz, CDCl_3_): δ = 7.84–7.69 (m, 2H), 7.57–7.42 (m, 5H), 7.42–7.35 (m, 2H), 6.45 (s, 1H), 2.45 (s, 3H), 2.35 (s, 3H) **^13^C NMR** (75.47 MHz, CDCl_3_): δ = 197.7, 194.3, 157.3, 151.4, 135.9, 132.9, 132.1, 129.3, 129.23, 129.19, 123.1, 106.5, 30.6, 26.0. **FT-IR** (thin layer): ν_max_ = 1725, 1697, 1488, 1413, 1363, 1296, 1091, 1049, 1019, 939, 821, 768, 691. **HR-MS** (ESI): *m/z* = 380.0770, cald. for C_18_H_16_ClN_3_O_3_+Na^+^: 380.0772.**(*E*)-3-(((4-methoxyphenyl)(phenyldiazenyl)methoxy)imino)pentane-2,4-dione**, **3bc,** was synthesized as a yellow solid (88%, purified by column chromatography with DCM as eluent). Mp = 69–70 °C **^1^H NMR** (300 MHz, CDCl_3_): δ = 7.81–7.69 (m, 2H), 7.51–7.38 (m, 5H), 6.98–6.89 (m, 2H), 6.44 (s, 1H), 3.81 (s, 3H), 2.45 (s, 3H), 2.35 (s, 3H). **^13^C NMR** (76 MHz, CDCl3): δ = 198.0, 194.5, 160.8, 157.0, 151.5, 131.8, 129.3, 129.2, 126.6, 123.0, 114.4, 107.2, 55.4, 30.6, 25.9. **FT-IR** (thin layer): ν_max_ = 1725, 1692, 1515, 1360, 1300, 1253, 1027, 951. **HR-MS** (ESI): *m/z* = 376.1261 cald. for C_19_H_19_N_3_O_4_+Na^+^ = 376.1268.**(*E*)-3-(((4-chlorophenyl)(methyldiazenyl)methoxy)imino)pentane-2,4-dione**, **3bd,** was synthesized as a yellow oil (68%, purified by column chromatography with DCM as eluent). **^1^H NMR** (300.13 MHz, DMSO-d_6_): δ = 7.53 (d, *J* = 8.6 Hz, 2H), 7.47 (d, *J* = 8.6 Hz, 1H), 6.36 (s, 1H), 3.85 (s, 3H), 2.35 (s, 3H), 2.32 (s, 3H). **^13^C NMR** (75.47 MHz, DMSO-d_6_): δ = 198.2, 193.7, 156.9, 134.5, 133.1, 129.5, 128.9, 104.5, 57.0, 30.1, 25.6. **FT-IR** (thin layer): ν_max_ = 1727, 1693, 1493, 1363, 1298, 1090, 977, 950. **HR-MS** (ESI): *m/z* = 318.0611, cald. for C_13_H_14_ClN_3_O_3_+Na^+^: 318.0616.**(*E*)-3-(((2-(phenyldiazenyl)propan-2-yl)oxy)imino)pentane-2,4-dione**, **3ca,** was synthesized as a yellow oil (85%, purified by column chromatography with DCM as eluent). **^1^H NMR** (300.13 MHz, CDCl_3_): δ = 7.76–7.67 (m, 2H), 7.52–7.43 (m, 3H), 2.44 (s, 3H), 2.35 (s, 3H), 1.62 (s, 6H). **^13^C NMR** (75.47 MHz, CDCl_3_): δ = 198.9, 194.8, 156.3, 151.6, 131.3, 129.2, 122.6, 104.7, 30.6, 25.8, 23.5. **FT-IR** (thin layer): ν_max_ =1726, 1690, 1384, 1303, 1196, 1173, 1145, 1070, 963, 767, 691. **HR-MS** (ESI): *m/z* = 298.1160, calcd. for C_14_H_17_N_3_O_3_+Na^+^: 298.1162.**(*E*)-3-(((4-(phenyldiazenyl)heptan-4-yl)oxy)imino)pentane-2,4-dione**, **3cb,** was synthesized as a yellow oil (73%, purified by column chromatography with DCM as eluent). **^1^H NMR** (300.13 MHz, CDCl_3_): δ = 7.75–7.63 (m, 2H), 7.54–7.41 (m, 3H), 2.44 (s, 3H), 2.34 (s, 3H), 2.15–1.91 (m, 4H), 1.55–1.19 (m, 4H), 0.90 (t, *J* = 7.3 Hz, 6H). **^13^C NMR** (75.47 MHz, CDCl_3_): δ = 199.0, 194.8, 156.2, 151.6, 131.2, 129.2, 122.5, 107.5, 37.2, 30.4, 25.8, 16.3, 14.6. **FT-IR** (thin layer): ν_max_ = 2964, 2934, 2875, 1726, 1690, 1363, 1303, 960, 768, 691. **HR-MS** (ESI): *m/z* = 354.1782, calcd. for C_18_H_25_N_3_O_3_+Na^+^: 354.1788.**(*E*)-3-(((6-(phenyldiazenyl)undecan-6-yl)oxy)imino)pentane-2,4-dione**, **3cc,** was synthesized as a yellow oil (70%, purified by column chromatography with DCM as eluent). **^1^H NMR** (300.13 MHz, CDCl_3_): δ = 7.73–7.64 (m, 2H), 7.53–7.42 (m, 3H), 2.44 (s, 3H), 2.34 (s, 3H), 2.13–1.93 (m, 4H), 1.49–1.18 (m, 12H), 0.86 (t, *J* = 6.8 Hz, 6H). **^13^C NMR** (75.47 MHz, CDCl_3_): δ = 198.9, 194.8, 156.3, 151.7, 131.1, 129.2, 122.5, 107.6, 34.8, 32.2, 30.4, 25.8, 22.5, 22.4, 14.1. **FT-IR** (thin layer): ν_max_ = 2957, 2932, 2870, 1727, 1692, 1363, 1301, 960, 767. **HR-MS** (ESI): *m/z* = 410.2402, calcd. For C_22_H_33_N_3_O_3_+Na^+^: 410.2414.**(*E*)-3-(((2-(phenyldiazenyl)hex-5-en-2-yl)oxy)imino)pentane-2,4-dione**, **3cd,** was synthesized as a slightly yellow viscous gum (79%, purified by column chromatography with PE/EA = 10/1 as eluent). **^1^H NMR** (300.13 MHz, CDCl_3_): δ = 7.76–7.66 (m, 2H), 7.54–7.44 (m, 3H), 5.93–5.62 (m, 1H), 5.21–4.77 (m, 2H), 2.45 (s, 3H), 2.35 (s, 3H), 2.27–1.98 (m, 4H), 1.63 (s, 3H). **^13^C NMR** (75.47 MHz, CDCl_3_): δ = 198.9, 194.8, 156.4, 151.6, 137.7, 131.4, 129.2, 122.6, 115.1, 106.0, 36.4, 30.5, 27.4, 25.9, 21.4. **FT-IR** (thin layer): ν_max_ = 1726, 1690, 1420, 1367, 1303, 982, 960, 826. **HR-MS** (ESI): *m/z* = 338.1475, calcd. for C_17_H_21_N_3_O_3_+Na^+^: 338.1475.**(*E*)-3-((1-(phenyldiazenyl)cyclobutoxy)imino)pentane-2,4-dione**, **3ce,** was synthesized as a slightly yellow viscous gum (89%, purified by column chromatography with DCM as eluent). **^1^H NMR** (300.13 MHz, CDCl_3_): δ = 7.81–7.70 (m, 2H), 7.56–7.42 (m, 3H), 2.69–2.50 (m, 4H), 2.46 (s, 3H), 2.35 (s, 3H), 2.09–1.84 (m, 2H).**^13^C NMR** (75.47 MHz, CDCl_3_): δ = 198.7, 194.7, 157.3, 151.7, 131.4, 129.2, 122.8, 105.2, 31.9, 30.7, 25.9, 12.0. **FT-IR** (thin layer): ν_max_ = 1727, 1690, 1364, 1304, 1251, 1143, 954, 768, 690. **HR-MS** (ESI): *m/z* = 310.1163, calcd. for C_15_H_17_N_3_O_3_+Na^+^: 310.1162.**(*E*)-3-(((1-(phenyldiazenyl)cyclopentyl)oxy)imino)pentane-2,4-dione**, **3cf,** was synthesized as a slightly yellow viscous gum (82%, purified by column chromatography with DCM as eluent). **^1^H NMR** (300.13 MHz, CDCl_3_): δ = 7.78–7.63 (m, 2H), 7.55–7.40 (m, 3H), 2.44 (s, 3H), 2.35 (s, 3H), 2.30–2.12 (m, 4H), 1.96–1.81 (m, 4H). **^13^C NMR** (75.47 MHz, CDCl_3_): δ = 198.8, 194.8, 156.8, 151.7, 131.2, 129.2, 122.6, 115.6, 36.4, 30.5, 25.9, 24.8. **FT-IR** (thin layer): ν_max_ = 2959, 1725, 1685, 1363, 1302, 1188, 959, 766, 690. **HR-MS** (ESI): *m/z* = 340.1059, calcd. for C_16_H_19_N_3_O_3_+K^+^: 340.1058.**(*E*)-3-(((1-(phenyldiazenyl)cyclohexyl)oxy)imino)pentane-2,4-dione**, **3cg,** was synthesized as a slightly yellow viscous gum (89%, purified by column chromatography with DCM as eluent). **^1^H NMR** (300.13 MHz, CDCl_3_): δ = 7.74–7.64 (m, 2H), 7.55–7.39 (m, 3H), 2.46 (s, 3H), 2.45 (s, 3H), 2.19–2.06 (m, 2H), 1.92–1.68 (m, 5H), 1.67–1.46 (m, 2H), 1.45–1.27 (m, 1H). **^13^C NMR** (75.47 MHz, CDCl_3_): δ = 198.9, 194.8, 156.6, 151.7, 131.2, 129.2, 122.6, 105.2, 32.1, 30.6, 25.9, 25.0, 21.9. **FT-IR** (thin layer): ν_max_ = 2938, 2863, 1727, 1689, 1599, 1450, 1420, 1363, 1304, 1275, 1256, 1195, 1159, 1146, 1069, 1023, 983, 960, 928, 911, 766, 691. **HR-MS** (ESI): *m/z* = 316.1654, calcd. for C_17_H_21_N_3_O_3_+H^+^: 316.1656.**(*E*)-3-((1-(phenyldiazenyl)ethoxy)imino)pentane-2,4-dione**, **3da,** was synthesized as a pale brown gum (74%, purified by column chromatography with DCM as eluent). **^1^H NMR** (300 MHz, CDCl_3_): δ = 7.79–7.69 (m, 2H), 7.54–7.44 (m, 3H), 5.67 (q, *J* = 6.3 Hz, 1H), 2.45 (s, 3H), 2.34 (s, 3H), 1.59 (d, *J* = 6.3 Hz, 3H). **^13^C NMR** (75 MHz, CDCl_3_) δ = 198.2, 194.5, 156.8, 151.5, 131.7, 129.3, 122.8, 103.5, 30.6, 25.9, 17.5. **FT-IR** (thin layer): ν_max_ = 1727, 1691, 1365, 1299, 1107, 1088, 1060, 965, 770, 691. **HR-MS** (ESI): *m/z* = 300.0733, calcd. for C_13_H_15_N_3_O_3_+K^+^: 300.0745.**(*E*)-3-((1-(phenyldiazenyl)propoxy)imino)pentane-2,4-dione**, **3db,** was synthesized as a pale yellow gum (59%, purified by column chromatography with DCM as eluent). **^1^H NMR** (300 MHz, CDCl_3_): δ = 7.83–7.64 (m, 2H), 7.60–7.39 (m, 3H), 5.55–5.41 (m, 1H), 2.45 (s, 3H), 2.33 (s, 3H), 2.15–1.88 (m, 2H), 1.05 (t, *J* = 7.5 Hz, 3H). **^13^C NMR** (75 MHz, CDCl_3_) δ = 198.2, 194.6, 156.9, 151.5, 131.6, 129.3, 122.8, 107.9, 30.6, 25.9, 25.3, 8.7. **FT-IR** (thin layer): ν_max_ = 1726, 1691, 1363, 1301, 1022, 988, 950, 769, 691. **HR-MS** (ESI): *m/z* = 298.1152, calcd. for C_14_H_17_N_3_O_3_+Na^+^: 298.1162.**(*E*)-3-((2-methyl-1-(phenyldiazenyl)propoxy)imino)pentane-2,4-dione**, **3dc,** was synthesized as a yellow gum (68%, purified by column chromatography with DCM as eluent). **^1^H NMR** (300 MHz, DMSO-*d*_6_): δ = 7.77–7.68 (m, 2H), 7.62–7.49 (m, 3H), 5.42 (d, *J* = 5.5 Hz, 1H), 2.40 (s, 3H), 2.39–2.28 (m, 1H), 2.26 (s, 3H), 0.99 (t, *J* = 7.5 Hz, 6H). **^13^C NMR** (75 MHz, DMSO-*d*_6_): δ = 198.3, 193.7, 156.7, 150.9, 131.8, 129.4, 122.3, 109.1, 31.1, 30.0, 25.5, 17.5, 16.6. **FT-IR** (thin layer): ν_max_ = 2970, 1726, 1691, 1364, 1299, 1020, 998, 959, 769, 691. **HR-MS** (ESI): *m/z* = 290.1500, calcd. For C_15_H_19_N_3_O_3_+H^+^: 290.1499.**(*E*)-3-((2,2-dimethyl-1-(phenyldiazenyl)propoxy)imino)pentane-2,4-dione**, **3dd,** was synthesized as a pale yellow gum (21%, purified by column chromatography with DCM as eluent). **^1^H NMR** (300 MHz, CDCl_3_): δ = 7.80–7.69 (m, 2H), 7.53–7.44 (m, 3H), 5.27 (s, 1H), 2.44 (s, 3H), 2.29 (s, 3H), 1.08 (s, 9H). **^13^C NMR** (75 MHz, CDCl_3_): δ = 198.1, 194.5, 156.8, 151.6, 131.6, 129.3, 122.9, 112.1, 36.0, 30.4, 25.7. **FT-IR** (thin layer): ν_max_ = 2973, 1727, 1693, 1365, 1300, 1021, 999, 959. **HR-MS** (ESI): *m/z* = 326.1474, calcd. for C_16_H_21_N_3_O_3_+Na^+^: 326.1475.**(*E*)-3-(((1-(phenyldiazenyl)hexyl)oxy)imino)pentane-2,4-dione**, **3de,** was synthesized as a yellow gum (70%, purified by column chromatography with DCM as eluent). **^1^H NMR** (300 MHz, CDCl_3_): δ = 7.80–7.67 (m, 2H), 7.55–7.42 (m, 3H), 5.55 (dd, *J* = 7.7, 5.1 Hz, 1H), 2.45 (s, 3H), 2.33 (s, 3H), 2.11–1.76 (m, 2H), 1.55–1.40 (m, 2H), 1.40–1.23 (m, 4H), 0.89 (t, *J* = 6.9 Hz, 3H). **^13^C NMR** (75 MHz, CDCl_3_): δ = 198.2, 194.6, 156.8, 151.5, 131.6, 129.2, 122.8, 107.0, 31.8, 31.6, 30.6, 25.9, 23.9, 22.5, 14.1. **FT-IR** (thin layer): ν_max_ = 2956, 2931, 1727, 1691, 1363, 1299, 964. **HR-MS** (ESI): *m/z* = 318.1810, calcd. for C_17_H_23_N_3_O_3_+H^+^: 318.1812.


**Experimental details for gram-scale synthesis of 3aa (Figure 3)**


Hydrazone **2aa** (4 mmol, 840 mg) was added to a stirred solution of diacetyliminoxyl **1** (8 mmol) in CH_2_Cl_2_ (200 mL) at room temperature. The reaction mixture was stirred at RT for 15 min, then rotary evaporated under a water-jet vacuum to an approximate volume of 40 mL. The reaction mixture was successively washed with 50 mL of saturated solution of NaHCO_3_, 50 mL of water, dried over MgSO_4_, and rotatory evaporated under a water-jet vacuum. The obtained C–O coupling product **3aa** (1.15 g, 3.41 mmol) was analytically pure, which was further confirmed by ^1^H and ^13^C NMR spectroscopy.


**Experimental details for one-pot procedure for the synthesis of 3ca (Figure 3)**


Phenylhydrazine (1 mmol, 108 mg) was dissolved in acetone (5 mL) and stirred at room temperature for 30 min. Then, the resulting solution was added dropwise to a stirred solution of diacetyliminoxyl **1** (2 mmol) in CH_2_Cl_2_ (50 mL). The obtained reaction mixture was stirred at RT for 15 min, and was then rotary evaporated under a water-jet vacuum. The C–O coupling product **3ca** was purified by column chromatography on silica gel as described in the experimental details of Figure 2.


**Experimental details for reaction of 1 with *N*,*N*-disubstituted hydrazone 2ao (Figure 4)**


Hydrazone **2ao** (1 mmol, 306 mg) was added to a stirred solution of diacetyliminoxyl **1** (2 mmol) in CH_2_Cl_2_ (50 mL) at room temperature. The reaction mixture was stirred at RT for 24 h and analyzed by ^1^H-NMR spectroscopy using 1,1,2,2-tetrachloroethane as an internal standard.


**Experimental details for TEMPO scavenging experiment (Figure 4).**


TEMPO (2 mmol, 312 mg) and hydrazone **2aa** (1 mmol, 210 mg) were added to a stirred solution of diacetyliminoxyl radical **2** (2 mmol) in DCM (50 mL). The reaction mixture was stirred for 15 min at room temperature and was then rotary evaporated under water-jet vacuum. Column chromatography on silica gel afforded **3aa** (310 mg, 0.92 mmol, 92%) and TEMPO (284 mg, 1.81 mmol, 91% recovery).

**Fungicidal activity tests (experimental details for** Table 2 and Table 3**).** The standard poison food technique [58,74,75,76,77,78] was used for fungicidal activity measurements against six phytopathogenic fungi of different taxonomic classes: *V.i.*—Venturia inaequalis MRA-16-2, *R.s.*—Rhizoctonia solani 100063, *F.o.*—Fusarium oxysporum FO-8, *F.m.*—Fusarium moniliforme 100146, *B.s*.—Bipolaris sorokiniana MRB(V)-1, *S.s*.—Sclerotinia sclerotiorum 100033. The strains used in this work were obtained from the collection of the All-Russian Research Institute for Phytopathology (B. Vyazemy, Moscow reg., Russia). The tested substances were dissolved in acetone (concentration 1 mg/mL) and introduced into liquid sugar-potato agar at 50–55 °C, so that the final substance concentration in the nutrient medium was 10 mg/L, and mixed thoroughly. Then, the agar containing the tested substance was poured into sterile Petri dishes. After the cooling of agar to room temperature, pieces of mycelium from the peripheral growth zone of a 3–5 day old culture of the fungus were transferred to test Petri dishes using a needle. A colony grown in the same medium without the addition of a fungicidal substance (same volume of acetone without any substance was added) was used as a control. The diameters of the formed fungal colonies were measured 72 h after inoculation. Each experiment was repeated 3 times, except for tests with *V.i*. culture which were conducted in 5 replicates. The suppression of mycelium growth in comparison with the control was calculated as ((Dc – Ds)/Dc) × 100%, where Dc is an average fungus colony diameter in control medium, and Ds is an average fungus colony diameter in the presence of the tested substance. Serial two-fold dilution experiments were conducted for EC_50_ determination.

**Computational details (Figure 5).** DFT calculations were conducted by the B97-3c composite method [35] including D3 dispersion correction [79,80], as implemented in the Orca 5.0.4 program [81]. The main conformers of diacetyliminoxyl **1** and hydrazone **2ca** were considered in all calculations. The presented results correspond to 218.15 K and 1 atm. See the Appendix A for the cartesian coordinates and energy values of optimized structures of **1a**, **1**, **2ca**, **A**, **B**, and transition states for path I and path II. Optimized geometries were visualized by the Avogadro 1.2 program [82]. 

## 5. Conclusions

In summary, we have disclosed the oxidative C–O coupling of hydrazones with diacetyliminoxyl as a ready-to-use radical reagent playing two roles: the role of a hydrogen atom acceptor and the role of a partner for the C–O coupling. The developed protocol is compatible with both aromatic and aliphatic keto- and aldehyde-derived hydrazones. Synthesized azo oxime ethers were discovered as a novel structural fungicide type with activity against phytopathogenic fungi that is comparable to the activity of commercial fungicides (triadimefon and kresoxim-methyl).

## Data Availability

Data are contained within the article and Appendix A.

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
