# Peer review of "C–O Coupling of Hydrazones with Diacetyliminoxyl Radical Leading to Azo Oxime Ethers—Novel Antifungal Agents"

_molecules, 2023, doi:10.3390/molecules28237863_

Round 1

Reviewer 1 Report

Comments and Suggestions for Authors

Budnikov and Krylov et al. reported ‘’ Radical ``click`` reaction: C-O coupling of hydrazines with diacetyliminoxyl radical leading to azo oxime ethers-compounds with unexpected antifungal activity’’.  This manuscript will be of great interest to many readers who are interested in learning about or starting the chemistry of diacetyliminoxyl radical.  In addition, the finding that the synthesized derivatives have antifungal activity is also an appreciated discovery.  

So, the reviewer recommends its publication of this work in ‘’Molecules’’ after minor revision as follows.

Additional comments.

A PDF is attached and should be corrected accordingly.

Author Response

Dear Reviewer,

Step-by-step answers are given below. Answers to all Reviewers is also attached as a PDF file.

Answer: We greatly thank the Reviewer for the high evaluation of our work. According to the Reviewer’s proposals in the attached PDF file, corresponding corrections have been made to the submitted manuscript.

Reviewer 2 Report

Comments and Suggestions for Authors

The work reports an oxidative C-O coupling of acetyliminoxyl radical with hydrazones, that affords azo oxime ethers, which were found to have unexpected and significant antifungal activity against different phytopathogenic fungi.

The methodology was applied to a wide number of hydrazones, derived from numerous, different ketones, but always with the same oxy radical. Although the synthesis of the radical requires toxic reagent (like Pb(OAc)4) and that is not very attractive, and the reaction seems to work only with one radical, the work might be interesting for the organic chemists community, since it opens the way to synthesize novel, functionalized molecules.

However, the authors should better describe the relevance of the new compounds, that is not very clear; are other analogous compounds known? And in which field? Could the authors give examples of important, structurally related, compounds?

Furthermore, the authors should give at least a few example of synthetic elaboration of the new synthesized compounds.

After the issues will be addressed the manuscript could be accepted for publication.

Author Response

Dear Reviewer,

Step-by-step answers are given below. Answers to all Reviewers is also attached as a PDF file.

Reviewer’s comment: The work reports an oxidative C-O coupling of acetyliminoxyl radical with hydrazones, that affords azo oxime ethers, which were found to have unexpected and significant antifungal activity against different phytopathogenic fungi.

The methodology was applied to a wide number of hydrazones, derived from numerous, different ketones, but always with the same oxy radical. Although the synthesis of the radical requires toxic reagent (like Pb(OAc)4) and that is not very attractive, and the reaction seems to work only with one radical, the work might be interesting for the organic chemists community, since it opens the way to synthesize novel, functionalized molecules.

Answer: Thank you for the positive assessment of the manuscript. Regarding the usage of Pb(OAc)4 we should note that it is the most convenient oxidant for the fast and almost quantitative preparation of diacetyliminoxyl as an individual reagent. The simple separation of diacetyliminoxyl from Pb compounds allows us to study the chemical properties of diacetyliminoxyl without possible intervention of metal ions or other oxidant-derived compounds. However, we agree that Pb(OAc)4 is not very attractive from the viewpoint of green chemistry principles and we will continue our research in the future to develop even more attractive methods for oxime radical generation.

Reviewer’s comment: However, the authors should better describe the relevance of the new compounds, that is not very clear; are other analogous compounds known? And in which field? Could the authors give examples of important, structurally related, compounds?

Answer: The most structurally relevant azo compounds are produced by oxidative functionalization of hydrazones:

Syntheses of these structures were described in the introduction:

Hydrazones are reported to undergo peroxidation by t-BuOOH in the presence of cobalt salen complexes with formation of geminal azoperoxides and geminal azoxyperoxides [14]. Unstable geminal azohydroperoxides [15] are formed as a result of hydrazone autoxidation by molecular oxygen [16–19]. In general, free-radical functionalization of hydrazones with formation of azocompounds is less developed compared to methods based on electrophilic attack of hydrazone carbon atom, such as Michael-type reactions [20–23], chlorination [24], alkoxylation or cyanation [25]. Geminal azoacetates are synthesized by oxidation of hydrazones with Pb(OAc)4 [26–28].

However, none of these compounds were known as fungicides. According to the reviewer query, we added comment to the introduction:

It should be noted that none of the products of oxidative functionalization of hydra-zones mentioned above were considered as fungicidal compounds.

Known azo-containing fungicidal agents are not structurally related to the azo oxime ethers synthesized in the present work. The corresponding comment was added to the section 3 (In vitro fungicidal activity of the synthesized azo compounds) after the sentence “The closest related fungicidal compounds are generally diaryl azo derivatives [60–62] and substituted oxime derivatives with a RO–N=C–N=N-Ar fragment at the oxime moiety [63].”:

“In contrast to these fungicides, azo oxime ethers reported in the present work contain tertiary C(sp3) atom at the azo group.”

Reviewer’s comment: Furthermore, the authors should give at least a few example of synthetic elaboration of the new synthesized compounds.

After the issues will be addressed the manuscript could be accepted for publication.

Answer: In the present work we focused our attention on the study of fungicidal activity of the synthesized products instead of their further synthetic transformations. The discovered pronounced activity (on the level of modern commercial fungicides) is remarkable and unexpected, and it determines the practical importance of the present work. However, we agree with the Reviewer’s note. Indeed, it is very important to study chemical properties and possible transformations of any novel compound type. In our preliminary experiments we revealed that N=N-C(sp3)-O-N=C moiety in the synthesized compounds is quite labile, for example, the C–O bond is cleaved under the reductive conditions (Pd/C, H2, 1 atm., rt) The chemical reactivity of this moiety is not very convenient if one wants to perform synthetic transformation with the retention of N-C(sp3)-O fragment, but this reactivity may be responsible for the useful fungicidal activity. We plan to study chemical and biological properties of azo oxime ethers in more detail in the future work.

Reviewer 3 Report

Comments and Suggestions for Authors

Krylov and coll. report the oxidative C–O coupling of hydrazones with diacetyliminoxyl. The method is straightforward with high functional compatibility. Moreover, some of the prepared compound presented interesting antifungal activity. Overall, the article is well-written and the experimental part is correctly described and complete. I recommend publication in Molecules after minor corrections noted below:

1) Title: the use of the term “click” should be justified or removed. The term “unexpected” should be also removed; otherwise, why testing them for antifungal activity?

2) The term “atom-economic” (summary, introduction and conclusion) should not be used for this reaction because two eq. of oxidant are used for one eq. of hydrazine, leading to the product and derivative 1-H as waste.

3) The result of Run 4 in Table 1 is not correct. According to the stoichiometry of the reaction, the yield of 3aa cannot be greater than 50%.

4) p.5: It is not clear how the test reaction with TEMPO helps for understanding the mechanism

5) Scheme 4, (1): the conversion of 2ao is of 30%, but to give what compound?

6) Part 3: please explain why these compounds were tested for antifungal activity

7) p.10, line 172: please correct, “3da” is written twice

8) p.11, lines 239-240: it is written that the “Yields were determined by 1H NMR using 1,1,2,2-tetrachloroethane as an internal standard”. However, in the following, all the yields are giver after column chromatography. So, what is the role of the internal standard here?

Author Response

Dear Reviewer,

Step-by-step answers are given below. Answers to all Reviewers is also attached as a PDF file.

Reviewer’s comment: Krylov and coll. report the oxidative C–O coupling of hydrazones with diacetyliminoxyl. The method is straightforward with high functional compatibility. Moreover, some of the prepared compound presented interesting antifungal activity. Overall, the article is well-written and the experimental part is correctly described and complete. I recommend publication in Molecules after minor corrections noted below.

Answer: We greatly thank the reviewer for the kind words about our work and for the critical analysis of the manuscript.

Reviewer’s comment: 1) Title: the use of the term “click” should be justified or removed. The term “unexpected” should be also removed; otherwise, why testing them for antifungal activity?

Answer: The manuscript title was corrected:

Old title: “Radical “click” reaction: C–O coupling of hydrazones with diacetyliminoxyl radical leading to azo oxime ethers — compounds with unexpected antifungal activity

Corrected title: “C–O coupling of hydrazones with diacetyliminoxyl radical leading to azo oxime ethers — novel antifungal agents

Reviewer’s comment: 2) The term “atom-economic” (summary, introduction and conclusion) should not be used for this reaction because two eq. of oxidant are used for one eq. of hydrazine, leading to the product and derivative 1-H as waste.

Answer: The term “atom-economic” was removed from the submitted manuscript.

Reviewer’s comment: 3) The result of Run 4 in Table 1 is not correct. According to the stoichiometry of the reaction, the yield of 3aa cannot be greater than 50%.

Answer: The yield is based on diacetyliminoxyl radical, which is the limiting reagent (in this run, hydrazone was taken in excess). According to the reaction stoichiometry, the theoretical maximum amount of the target product for 2 mmol of diacetyliminoxyl is 1 mmol, thus 1 mmol of the target product corresponds to the 100% yield. Corresponding note was added to the Table 1:

b Yields are based on 1 because 2aa is used in excess, according to the reaction stoichiometry, 100% corresponds to 1 mmol of 3aa or 1-H

Reviewer’s comment: 4) p.5: It is not clear how the test reaction with TEMPO helps for understanding the mechanism

Answer: An experiment with TEMPO is a typical control reaction which is usually employed to intercept possible C-centered radical intermediates. Comment was added:

“An experiment with TEMPO (Scheme 4, eq. 2) is a typical control reaction which is usually employed to intercept possible C-centered radical intermediates.”

No TEMPO-containing product was detected in our case. However, this fact does not exclude the formation of C-centered radical from hydrazone (for example, Org. Chem. Front., 2023, 10, 388-398, 10.1039/D2QO01649D).

Reviewer’s comment: 5) Scheme 4, (1): the conversion of 2ao is of 30%, but to give what compound?

Answer: A complex mixture of products was observed by 1H NMR. Unfortunately, no major product was identified.

Reviewer’s comment: 6) Part 3: please explain why these compounds were tested for antifungal activity

Answer: As a part of our ongoing program aimed at the development of new crop protection compounds, we systematically test novel compounds for fungicidal activity against phytopathogenic fungi. Special attention is devoted to unusual compound families which we make available by innovative synthetic methods developed in our group. Previously, we developed synthetic methods towards novel fungicidal compounds of other structural groups: pyrazolone derivatives (Chemistry A European J., 2019, 25, 5922–5933, J. Agric. Food Chem., 2022, 70, 4572−4581, Agrochemicals 2023, 2, 34-46), organic peroxides (Chem. Eur. J. 2020, 26, 4734, Agrochemicals 2023, 2, 355), thicyanates (Org. Biomol. Chem. 2023, 21, 3615, Chem. Het. Comp. 2021, 57, 531), and some tetrahydroquinoline derivatives (Adv. Synth. Catal. 2022, 364, 1098). Therefore, products 3 were tested for fungicidal activity against various phytopathogenic fungi from different taxonomic classes in the frame of continuous screening. To the best of our knowledge, no structurally relevant compounds with R-N=N-C(sp3)-O-N=CR’2 motif were known previously as fungicides.

Reviewer’s comment: 7) p.10, line 172: please correct, “3da” is written twice

Answer: Thank you for this note. Numbering was corrected:

Old text: “As can be seen from Table 2, compounds 3da and 3da exhibit the greatest activity against phytopathogenic fungi.”

Corrected text: “As can be seen from Table 2, compounds 3da and 3dc exhibit the greatest activity against phytopathogenic fungi.”

Reviewer’s comment: 8) p.11, lines 239-240: it is written that the “Yields were determined by 1H NMR using 1,1,2,2-tetrachloroethane as an internal standard”. However, in the following, all the yields are giver after column chromatography. So, what is the role of the internal standard here?

Answer: Thank you for this essential question. The internal standard was used for yield determination during optimization because it is quick and reliable. In addition, it was used to ensure that isolated yield correlates well with the actual yield from the reaction (for example, that there is no decomposition during column chromatography on silica gel, which can lead to a subsequent loss of the isolated yield). In the Scheme 2 the scope of synthetic applicability of the developed method is demonstrated. Isolated yields are valued higher in this case, because demonstrate actual amount of the target product which can be isolated after purification.
